# Heterogeneity of Patient-Derived Acute Myeloid Leukemia Cells Subjected to SYK In Vitro Inhibition

**DOI:** 10.3390/ijms232314706

**Published:** 2022-11-25

**Authors:** Marte Karen Brattås, Anette Lodvir Hemsing, Kristin Paulsen Rye, Kimberley Joanne Hatfield, Håkon Reikvam

**Affiliations:** 1Department of Clinical Science, University of Bergen, 5021 Bergen, Norway; 2Department of Medicine, Haukeland University Hospital, 5021 Bergen, Norway; 3Department of Immunology and Transfusion Medicine, Haukeland University Hospital, 5021 Bergen, Norway

**Keywords:** acute myeloid leukemia (AML), apoptosis, cytokines, proliferation, spleen tyrosine kinase (SYK)

## Abstract

Acute myeloid leukemia (AML) is an aggressive hematological malignancy with a dismal prognosis. The cytoplasmic spleen tyrosine kinase (SYK) is highly expressed by hematopoietic cells and has emerged as a potential therapeutic target. In this study, we evaluated the in vitro antileukemic effects of five SYK inhibitors, fostamatinib, entospletinib, cerdulatinib, TAK-659, and RO9021, in a consecutive AML patient cohort. All inhibitors demonstrated a concentration-dependent antiproliferative effect, although there was considerable heterogeneity among patients. For fostamatinib and TAK-659, the antiproliferative effects were significantly higher in *FLT3* mutated patients compared to nonmutated patients. Fostamatinib, entospletinib, TAK-659, and RO9021 induced significant apoptosis in primary AML cells, although the proapoptotic effects of the SYK inhibitors were less pronounced than the antiproliferative effects. Finally, most of the SYK inhibitors caused a significant decrease in the release of cytokines and chemokines from primary AML cells, indicating a potent inhibitory effect on the release of these leukemic signaling molecules. We concluded that the SYK inhibitors had antileukemic effects in AML, although larger studies are strongly needed to identify which patient subsets will benefit most from such a treatment.

## 1. Introduction

Acute myeloid leukemia (AML), the most common type of adult acute leukemia, is characterized by rapid cellular proliferation, an aggressive clinical course, and a high mortality rate [1]. Despite the increasing knowledge about the disease, AML treatment remains challenging owing to the disease’s complex biology, heterogeneity, and the coexistence of multiple genetically aberrant clones within the same patient [2]. Intensive “3 + 7” chemotherapy with anthracycline and cytarabine (AraC) remains the standard initial therapy [1]. Unfortunately, most patients eventually relapse, and the 5-year survival for AML patients remains unsatisfactory at approximately 40% in younger patients for the disease group as a whole [3]. Because of the risk of treatment-related mortality, this regimen is only offered to fit and younger patients below 65–70 years; furthermore, treatment options for older patients are limited, and the life expectancy for these patients is often short. Hence, new therapeutic approaches for AML are highly desired.

Spleen tyrosine kinase (SYK) is a cytoplasmic nonreceptor tyrosine kinase known to have oncogenic properties in AML [2,4,5,6,7]. It is highly expressed in hematopoietic cells and is considered a promising nonmutated therapeutic target in hematological malignancies, including AML [4,7]. High SYK activity in AML cells is associated with an unfavorable prognosis independent of age, cytogenetics, and leukocyte count [6,7,8]. Studies show that SYK is involved in the signaling pathways that drive AML [4,8,9,10,11,12,13,14,15,16], and the cascade of FMS-related tyrosine kinase 3 (FLT3) signaling are of special interest since approximately 1/3 of AML patients have a mutation in the *FLT3* gene [7].

Several SYK inhibitors have recently been developed, and bioavailable SYK inhibitors have entered clinical trials for patients with AML [17,18,19], with promising results [17,18,19,20,21]. Previous in vitro studies in mouse models have resulted in the impaired growth of AML cells [2,10]; however, the underlying molecular events of SYK signaling have not been investigated in a large context [4]. In the present study, we explored the in vitro effects of five different SYK inhibitors on the viability and proliferation of primary leukemia cells derived from a large group of consecutive AML patients. To the best of our knowledge, examining the effect of different SYK inhibitors simultaneously by targeting primary AML cells has not yet been reported. Our aim was to investigate and compare the effects of SYK inhibition on leukemia cells and to evaluate any associations with AML patient characteristics, e.g., mutational status or cytogenetics.

## 2. Results

### 2.1. Dose–Response Studies of AML Cell Proliferation after Exposure to SYK Inhibitors

We tested the ability of primary AML cells to survive and still proliferate seven days after treatment with the five SYK inhibitors: fostamatinib, entospletinib, cerdulatinib, TAK-659, and RO9021. For the initial dose–response curve, we tested seven concentrations with a broad range on a cohort of seven randomly selected AML patients (Table 1). These concentrations were chosen based on the literature search of these inhibitors used in previous in vitro studies on hematological malignancies [22,23,24,25,26,27,28]. AML cell proliferation was assessed using the [^3^H]-thymidine incorporation assay. Detectable proliferation, defined as >1000 counts per minute (cpm), was found in all untreated patient samples. Cerdulatinib (0.01–10 µM) and TAK-659 (0.005–5 µM) showed a strong antiproliferative effect in all seven patient samples. A more divergent antiproliferative effect was demonstrated after inhibition by fostamatinib (0.01–5 µM), entospletinib (0.01–150 µM), and RO9021 (0.01–10 µM) in the same seven patient samples. Overall, all five SYK inhibitors demonstrated a concentration-dependent antiproliferative effect in all seven patient samples, and the results are presented in Figure 1A.

### 2.2. SYK Inhibition Demonstrated the Antiproliferative Effect in a Larger AML Patient Cohort

We next studied the antiproliferative effect of these five SYK inhibitors in a large cohort of 68 consecutive AML patients for two selected drug concentrations. We aimed to use concentrations resulting in a reduced AML cell proliferation of approximately 30% and 70% compared to the untreated controls. Therefore, based on the results of the initial dose–response study (Figure 1A), the following concentrations were chosen for further studies: 1 µM and 0.1 µM fostamatinib, 1 µM and 10 µM entospletinib, 0.01 µM and 0.05 µM cerdulatinib, 0.05 µM and 0.5 µM TAK-659, and 0.5 µM and 5 µM RO9021. Detectable proliferation, defined as a [^3^H]-thymidine incorporation of >1000 cpm was found for 59 (87%) of the 68 patient samples. As seen in Figure 1B, all SYK inhibitors demonstrated a wide range of antiproliferative effects. Table 1 shows the overall median AML cell proliferation of the 59 patient samples with detectable proliferation for the two tested concentrations of each inhibitor.

Overall, treatment with the SYK inhibitors resulted in a significant antiproliferative effect compared to the untreated controls (*p*-values in Figure 1B; Wilcoxon signed-rank test) in both concentrations of all inhibitors.

### 2.3. Antiproliferative Effects of SYK Inhibition Varied between AML Patients

Next, we compared the sensitivity toward SYK inhibition of the 59 patient samples with detectable proliferation (>1000 cpm) by assessing the percent proliferation compared to the controls (set to 100%) using the [^3^H]-thymidine incorporation assay. As illustrated in Figure 2, we identified a high degree of heterogeneity between the antiproliferative effects of patient samples after treatment with the different SYK inhibitors. For some patients, AML cell proliferation was inhibited completely with SYK inhibition, while other patients had no effect with the same concentration. Two patient cohorts were identified by assessing the inhibitors’ overall high and low antiproliferative effects calculated by the 50th percentile. Fisher’s exact test was used to evaluate if the observed antiproliferative effects after treatment with all five SYK inhibitors were associated with known disease entities or etiological features, such as the French–American–British (FAB) classification system, cytogenetics, an *FLT3* mutation, an *NPM1* mutation, CD34 expression, gender, or age. No significant associations were found, indicating that SYK inhibitors have antiproliferative effects in these different AML patient subgroups.

### 2.4. Antiproliferative Effects of Fostamatinib and TAK-659 Were Significantly Higher in FLT3 Mutated AML Patients

We evaluated the inhibitory effect on AML cell proliferation of each SYK inhibitor alone and with the association of the prognostic mutations *NPM1* and *FLT3*. The Mann–Whitney U test was used to compare groups of patients with or without mutations. A significant difference in the antiproliferative effects of SYK inhibitors between *FLT3* mutated patients and patients without *FLT3* mutations was detected in the highest concentrations of fostamatinib and TAK-659 (Mann–Whitney U test; *p* = 0.0233 and *p* = 0.0132, respectively (Figure 3). For the other SYK inhibitors, a trend toward higher antiproliferative effects in *FLT3* mutated patients was detected, although this did not reach statistical significane.

We also evaluated any associations between the *NPM1* mutational status and the proliferative responses after drug treatment, though no significant associations were found for any of the SYK inhibitors. Furthermore, using Fisher’s exact test, we did not find any significant correlations between a high antiproliferative effect and the presence of an *FLT3* mutation or an *NPM1* mutation when testing the highest concentration of each SYK inhibitor.

### 2.5. SYK Inhibitors Decreased AML Cell Viability and Induced Apoptosis

We examined the proapoptotic effects of the five SYK inhibitors using the annexin-V/propidium iodine (PI) assay of flow cytometry. Both concentrations of the inhibitors that were tested in the AML cell proliferation assay were tested in the apoptosis flow cytometric assay. The gating strategy used to define the percentages of the viable (annexin-V negative/PI negative), apoptotic (annexin-V positive/PI negative), and necrotic (annexin-V positive/PI positive) cells is shown for one representative patient sample in Figure 4. Of the 68 patient samples analyzed, 54 samples had more than 10,000 events as well as 15% of viable cells in the untreated control cultures, and these patient samples were included in the statistical analysis.

As illustrated in Figure 5, treatment with the lowest concentration of each SYK inhibitor resulted in relatively small changes in AML cell viability in the 54 patients. When comparing the effect of the treatment with the SYK inhibitors on AML cell viability to the untreated control, we found a significant decrease in viability after treatment with the highest concentrations of fostamatinib, entospletinib, TAK-659, and RO9021 (Wilcoxon signed-rank test; *p* = 0.0025, 0.0001, 0.0006, and 0.0001, respectively). The leukemic cell viability among all patients was heterogeneous; the median viability after 48 h for the untreated controls was 44.3% (range 10.0–96.4%). Overall, the AML cells showed the greatest sensitivity toward 5µM of RO9021 for which the viability ranged from 34.8–47.3%.

When evaluating the proapoptotic effects of the SYK inhibitors on primary AML cells, we found that the percentage of apoptotic cells after 48 h was relatively high in both the untreated control cultures and the cultures treated with the SYK inhibitors. Still, the highest concentrations of fostamatinib, entospletinib, TAK-659, and RO9021 induced further significant proapoptotic effects (Wilcoxon signed-rank test; *p* = 0.0013, 0.0001, 0.0001, and 0.0001, respectively). We analyzed if there were any differences between the proapoptotic effects seen after the treatment with the SYK inhibitors and the *FLT3* mutation status for the patients, but no significant differences were found (Mann–Whitney U test).

### 2.6. SYK Inhibition Resulted in Reduced Release of Cytokines

The aberrant release of cytokines and other mediator molecules is a hallmark of AML. Therefore, the effect of SYK inhibition on the AML cell cytokine release for 13 unselected patients was examined. A total of 18 mediators and cytokines were examined, and they were selected because of their common detectable release by AML cells [29,30] (Table 2). Primary AML cells from 13 patients were cultivated for 48 h with or without SYK inhibitors before the concentrations of the mediators and cytokines were determined in the supernatants. Table 2 shows the primary release of the mediators measured in the untreated AML cell cultures.

All cytokines except CCL2, CXCL1, and granulocyte colony stimulating factor (G-CSF) had detectable concentrations for at least 9 of the 13 patients in the untreated AML cultures. All the patients had detectable proliferations (>1000 cpm) and more than 5% of viable cells in the control culture after 48 h of incubation. The release profiles showed a high degree of heterogeneity between patients and were similar to results reported in previous studies [30,31,32].

We further investigated the effects of SYK inhibition on the mediator release by AML cells. The highest concentrations of the inhibitors tested in the proliferation and apoptosis/necrosis assays were used to investigate the effects of the 48-h treatment of the SYK inhibitors on the AML cell mediator release. Figure 6 shows the median level of each cytokine, which was presented as a ratio of the treated cultures compared to the untreated control cultures.

SYK inhibition caused divergent effects on the release of cytokine mediators by AML primary cells, and we found a heterogenous effect among the different inhibitors. We found that most of the SYK inhibitors caused a significant reduction in the cytokine release for most of the chemokines (CCL2, CCL3, CCL4, CCL5, CXCL1, CXCL5, and CXCL10), but weak effects were seen in the release of growth factors (G-CSF, hepatocyte growth factor (HGF), and tumor necrosis factor alfa (TNF-α)) and the regulators of the protease system (Cystatin C, Serpin-E1, and MMP-2) (Figure 6B). TAK-659 showed an overall strong inhibitory effect on the release of all cytokines; however, the effects of the treatment were only significant for the levels of CCL2, CCL3, CCL4, CCL5, CXCL5, CXCL10, IL-8, IL-6, TNF-α, and matrix metalloproteinase (MMP-2) (Wilcoxon signed-rank test). By contrast, inhibition with fostamatinib increased the release of the cytokines CXCL1, IL-1Ra, IL-1β, G-CSF, and MMP-1. The mediator profile of IL-6 seemed relatively unchanged in the absence or presence of fostamatinib, entospletinib, and cerdulatinib inhibition. 

We finally performed an unsupervised hierarchical cluster analysis with a heatmap based on the effect of the cytokine release in the 13 AML patients with the five different SYK inhibitors compared to the untreated controls (Figure 7). The analysis demonstrated that the different SYK inhibitors mostly had the same effect on the release of single cytokines in the same patient. Taken together, this indicated that the SYK inhibitors tended to have the same effect on the cytokine release and that patients vulnerable to SYK inhibition tended to have decreased levels of the whole cytokine network with SYK inhibition in general.

## 3. Discussion

The SYK signaling pathway is implicated in leukemogenesis [4,8,9,10,11,12] and has emerged as a potential therapeutic target in AML [4,7]. Several SYK inhibitors have been developed; however, their pharmacological properties in a heterogeneous disease such as AML are only partially known. In the present study, we therefore investigated primary AML cells derived from a large group of unselected patients to study the in vitro effect of five different SYK inhibitors. The concentrations of the various drugs used in our present experiments were selected based on observations from previous studies as were our current pilot dose–response experiments in the proliferation assay with primary human AML cells.

Our results demonstrated that all the SYK inhibitors had concentration-dependent antiproliferative effects in primary AML cells (Figure 1A). Furthermore, since AML is a heterogenous disease, we hypothesized that differences in antiproliferative effects could be observed between different AML patients. The antiproliferative effects of all the inhibitors were assessed using two concentrations (Figure 1B) chosen based on the breaking point of the initial titration curve (Figure 1A). Each of the five SYK inhibitors demonstrated antiproliferative effects, and the highest inhibitory effect on proliferation was observed in the highest concentration tested (Figure 1B). By dividing the patient cohort into patients with strong inhibitory effects and low inhibitory effects toward the SYK inhibitors, we searched for an association between the disease entity and the inhibitory effect. However, no clear association between these disease features and the antiproliferative effect of the SYK inhibitors was detected, indicating that SYK inhibition could demonstrate antileukemic effects across different AML entities. Previous studies have also shown that SYK inhibitors affect different AML entities [33] and have clinical relevance given the heterogeneity in the AML patient population as a whole.

However, since SYK inhibition has demonstrated to be of special interest in *FLT3* mutated patients [9,34,35,36], demonstrating significant antileukemic effects, we compared the antiproliferative effects in patients harboring either *FLT3* wild type or *FLT3* mutations. We discovered a trend toward a stronger antiproliferative effect on primary AML cells with the SYK inhibitors in patients with *FLT3*; however, only for the highest concentrations of fostamatinib (1.0 µM) and TAK-659 (0.5 µM) did the difference in antiproliferative effects reach statistical significance (Figure 3). This could reflect the pharmacological properties of the different SYK inhibitors as fostamatinib is a competitive SYK/FLT3 inhibitor, and TAK-659 has also demonstrated potent inhibitory pharmacological effects against the *FLT3*-ITD dependent cell lines MV4-11 and MOLM-13 [37].

When evaluating the relationship between the high overall antiproliferative effects of the SYK inhibitors and *FLT3* mutations, we found no significant associations. However, only 13 patients with *FLT3* mutations were included in the patient cohort with a high antiproliferative effect, so the number of patients is likely too small to find a significant *FLT3*-associated difference between the two cohorts. It would therefore be of interest to evaluate a larger group of *FLT3* mutated patients.

The presence of *FLT3* mutations in AML patients has traditionally been associated with an inferior prognosis, although recent advances, including modifying the impact of FLT3 inhibitors, increasing the use of minimal residual disease monitoring, and better prognoses with allogeneic stem cell transplantation, have improved the prognosis of this subgroup of AML patients which are currently classified in the intermediate prognosis group [1]. However, resistance to FLT3 inhibitors and complications related to allogenic transplants still influence the treatment outcome of this patient subgroup. SYK activation has also been previously linked to *FLT3* mutated cases in AML [8], and our present data supported that SYK inhibition could be especially valuable for *FLT3* mutated patients. Furthermore, if a combination of more potent *FLT3* inhibitors and SYK inhibitors could have synergistic pharmacological properties in *FLT3* mutated AML cases, it remains unanswered and must be explored in further studies. Finally, we also used the FLT3 ligand in our proliferation assay, as the FLT3 ligand has been demonstrated to be important for AML cell *proliferation* in vitro [38]. However, the FLT3 ligand also exists in the normal and leukemic bone marrow [39,40], and, hence, our in vitro conditions were believed to reflect the in vivo conditions by introducing the FLT3 ligand to influence the leukemic cells. If the FLT3 ligand in the culture media was minimizing or masking an increased effect of the SYK inhibitors and if this has a potential transferable effect for in vivo studies, however, is unpredictable. Finally, the effect of the FLT3 ligand in the bone marrow microenvironment could be both concentration- and context-dependent. Hence, the culminating pharmacological effect of the SYK inhibitors in AML patients was apparently dependent on a number of inter- and intraindividual patient characteristics.

Previous studies have demonstrated that AML cell viability shows a wide variation among patients after in vitro incubation due to spontaneous or stress-induced apoptosis [41]. We found that SYK inhibition also had a proapoptotic effect in our in vitro study with primary AML cells. The proapoptotic effect also proved to be concentration-dependent for the different SYK inhibitors (Figure 4 and Figure 5). However, the proapoptotic effect seemed to be less prominent than the antiproliferative effect (Figure 4 and Figure 5), indicating that the antiproliferative effect was the most important effect for the antileukemic effects of the SYK inhibitors. Cerdulatinib especially had only a minor effect on the AML cell viability and the induction of apoptosis. This could reflect that the concentrations of certain inhibitors were too low for the aim as described previously and illustrated in Table 1. However, these findings were also consistent with previous in vitro studies on primary multiple myeloma cells, also indicating that the antiproliferative effect is more pronounced than the proapoptotic [42].

We also analyzed the constitutive release profile of cytokines and antiregulatory mediators commonly released by primary human AML cells, as this is a key feature of malignant hematological cells [43]. The SYK inhibitors in general demonstrated an inhibitory effect on the cytokine release for most of the cytokines evaluated (Figure 6). However, we observed a wide variation in the pharmacological effects on the constitutive mediator release between patients, and the inhibitory effects on the cytokine release were used to further study the different patients’ interindividual variations (Figure 7). We found that the five different SYK inhibitors mainly demonstrated the same effects on the cytokine release within the same patient. Interestingly, we also found that different SYK inhibitors demonstrated a trend in altering the levels of all cytokines evaluated within the same AML patient (Figure 7). These results indicated that the different SYK inhibitors had approximately the same effects in the same patient. A reduced cytokine release could reflect and increase the apoptotic effect of SYK inhibition; however, as demonstrated, the proapoptotic effect of SYK inhibition seemed limited (Figure 4 and Figure 5). Hence, we concluded that the reduced cytokine levels (Figure 6) could not be explained by the proapoptotic effects and probably reflected a general reduction in cytokine secretion induced by inhibiting the SYK-signaling pathway. The finding of an inhibitory effect on the cytokine release is also found in previous studies [44] and is recently exemplified by the use of SYK inhibitors for the treatment of cytokine storms associated with severe COVID-19 infections [45].

The constitutive activation and phosphorylation of SYK are reported in AML, and this activation seems to be independent of the driving oncogene and is probably caused by the tonic activation of cell surface receptors by cytokines or other mediators derived from the neighboring cell [2]. SYK mRNA is widely expressed in AML patients [11], and SYK expression at the protein level has been detected for more than 90% of patients. Furthermore, high activation has previously been reported to be associated with an adverse prognosis independent of age, karyotype, and peripheral blood blast count [8]. SYK and related network proteins also seem to be significantly higher-expressed in patients relapsing after first-line treatment compared to relapse-free patients [46]. Our present data supported that inhibition of SYK has an antileukemic effect which is effective across different disease entities and is demonstrated by effects in different cytogenetic and molecular genetic subgroups. The fact that some patients seem to have a lesser antileukemic effect can be caused by several factors, but both the development of resistance and the concentration variations of the different inhibitors may play a role.

Fostamatinib was one of the first identified SYK inhibitors [20]; however, more recently discovered SYK inhibitors have potencies that are considerably better than fostamatinib, and the development of SYK inhibitors for clinical use and entering them in clinical trials have been desired. Our study supported that the different SYK inhibitors often have a similar effect in the same patient. Whether some SYK inhibitors are more potent and may be more beneficial to use in certain subgroups of patients must be further studied in preclinical and clinical studies.

Recent years have seen a significant increase in the treatment armamentarium for use in AML therapy, which has probably also resulted in an improvement for this patient group [47]. These pharmacological approaches include the BCL-2 inhibitor venetoclax [48], the CD33 monoclonal antibody gemtuzumab ozogamicin [49], and the hedgehog inhibitor glasdegib [50], in addition to agent-specific inhibition ofrecurrent AML mutated genes, namely FLT3 inhibitors, such as midostaurin [51], gilteritinib [52], and quizartinib [53], as well as the IDH1 and IDH2 inhibitors ivosidenib and enasidenib [54,55], respectively. Hence, treatment physicians have more to offer AML patients which also include both patients with refractory or relapse disease and elderly and comorbid patients that are not candidates for intensive therapy. If SYK inhibitors in the future could also enter this pharmacological armamentarium for AML remains an unanswered question. However, our data supported an effect of SYK inhibition particularly in *FLT3* mutated cases, and special TAK-659, which has inhibitory effects against both FLT3 and SYK, seems promising for use in hematological malignancies [18]. The approaches of dual SYK and FLT3 inhibition was further supported by preclinical data, indicating that SYK is a critical regulator of FLT3 in AML [9].

An important question regarding the pharmacological approaches in hematological malignancies is: what are the potential differences between the SYK inhibitors in leukemic cells and in normal hematopoietic cells? We did not particularly address this in our present study. However, the experience regarding the SYK inhibitors in the clinical trials for hematological malignancies and the clinical practice for immune thrombocytopenia (ITP) is that the hematological toxicity, i.e., bone marrow suppression, is quite low [5,17,19,21,56,57,58,59]. Furthermore, since the SYK pathway seems to be upregulated in AML cells [4], it is likely that SYK inhibitors, like most other tyrosine kinase inhibitors, have limited effects on normal hematopoietic cells and, hence, hematological toxicity. As this study was performed to reveal the differences in the pharmacological properties within the AML patient population, we did not address the effects of SYK inhibition in normal hematopoietic cells nor in other hematological malignancies.

The precise mechanisms of action and the effect of SYK inhibition in the leukemic bone marrow microenvironment must be explored in future preclinical and clinical models. Finally, as the understanding of the hierarchy framework and heterogeneity in the AML disease biology is in continuous development, the complexity of the composition of the disease is still evolving [60]. Given this complexity, the question regarding how subdivided AML can be arises, resulting in the conclusion that the number of patients needed to detect effects on disease sublevels must be very high [61]. In the present study we detected small, although significant, differences between disease entities although larger studies must be performed to detect the future association between the disease biology and the effects of SYK inhibitors.

Our study had some limitations. Examining a large consecutive AML patient cohort involves challenges regarding a complete mapping of the heterogeneity in the AML patient population. New genetic mutations, in addition to *FLT3* and *NPM1*, have now also been established in the pathogenesis of AML and are included in new prognostication tools [1]. Larger studies are needed to fully address the question of the effects of SYK inhibition in different subgroups. The transferability from in vitro effects to real clinical patient effects can also be a difficult estimate. However, our results suggested that further studies should be performed to assess the potential of exploiting SYK inhibition as an attractive therapeutic strategy in human AML.

## 4. Materials and Methods

### 4.1. Patients and Primary Human AML Cells

The study was conducted in accordance with the Declaration of Helsinki, and the collection and use of samples for this study were approved by the regional committees for medical and health research ethics (REK) both for biobanking and in vitro experimental research (REK 1750/2015 and REK 480847/2022). Registration of collected samples was also approved by the Norwegian Data Protection Authority (reference 02/1118-5).

Peripheral blood mononuclear cells were obtained from AML patients at Haukeland University Hospital at diagnosis after written informed consent was obtained. AML cells were isolated from primary samples using density gradient separation (Lymphoprep^TM^, Alere Technologies, Oslo, Norway) and contained at least 95% leukemic blasts before being cryopreserved in RPMI-1640 (Sigma-Aldrich, St. Louis, MO, USA) with 10% dimethyl sulfoxide (DMSO) (Merck KGaA, Gernsheim, Germany) and 20% heat-inactivated fetal bovine serum (FBS) (BioWest, Nuaille, France). All samples were stored in liquid nitrogen until thawed and used in experiments. In our study, cells were derived from 68 consecutive AML patients, and their main characteristics are given in Table 3.

### 4.2. Reagents

The SYK inhibitors used in all experiments were fostamatinib disodium (MedChemExpress, (Monmouth Junction, NJ, USA)), entospletinib (MedChemExpress, (Monmouth Junction, NJ, USA)), cerdulatinib (Selleckchem, Houston, TX, USA), TAK-659 (Selleckchem, Houston, TX, USA), and RO9021 (Selleckchem, Houston, TX, USA), and their main pharmacological properties are listed in Table 4. A stock solution was prepared according to the manufacturer’s instructions before being aliquoted in RPMI-1640 or StemSpan SFEM^TM^ medium (Stem Cell Technologies, Vancouver, BC, Canada) and being stored at −80 °C. Aliquots were thawed only once before being used in experiments.

StemSpan SFEM^TM^ medium (Stemcell Technologies, Vancouver, BC, Canada) supplemented with exogenous G-CSF, stem cell factor (SCF), and FLT3 ligand (FLT3-L) (all from Peprotech, London, UK) was used as a cell culture medium in all experiments. Cytokines were used at a final concentration of 20 ng/mL.

### 4.3. AML Cell Proliferation Assay

AML cells were seeded in triplicates at 50,000 cells/well in 100 μL of StemSpan SFEM^TM^ supplemented with G-CSF, SCF, and FLT3 ligand in flat-bottomed 96-well plates.

In treated cultures, 100 μL of indicated concentrations of each of the five SYK inhibitors was added in StemSpan SFEM^TM^ medium per well, and 100 μL of StemSpan SFEM^TM^ medium was added to the treatment-free control cultures. The cultures were incubated at 37 °C in a humidified atmosphere for 6 days before 20 μL [^3^H]-thymidine (TRA 310; Perkin Elmer, NET027A005MC, Amersham International, Amersham, UK) per well was added. After an additional 20–22 h, cultures were harvested, and nuclear incorporation was measured using the [^3^H]-thymidine incorporation assay. Detectable proliferation was defined as >1000 cpm, and proliferation was estimated by comparing percent proliferation in treated cultures to treatment-free control cultures.

### 4.4. Apoptosis Assay

AML cells were seeded at 1 × 10^6^ cells/mL/well in 24-well plates with 500 μL of StemSpan SFEM^TM^ medium supplemented with G-CSF, SCF, and FLT3-L. In treated cultures, 500 μL of indicated concentrations of SYK inhibitors per well were added. 500 μL StemSpan SFEM^TM^ medium was added to treatment-free control cultures. After 48 h, the AML cells were washed twice with cold PBS. The cell pellets were resuspended in binding buffer and double stained with annexin V and PI in the dark for 15 min. Cell viability, apoptosis, and necrosis were analyzed with Pacific Blue^TM^ Annexin V Apoptosis Detection Kit with PI (BioLegend, San Diego, CA, USA), staining according to the manufacturer’s protocol. Unstained and single-stained control cells were included in each experiment. Measurements were performed using flow cytometry FACS Verse (BD FACSVerse 8 color flow) and the FlowJo 10.0.7 software (Tree Star, Inc., Ashland, OR, USA) [62]. A total of 10,000 events were collected and analyzed per sample. Doublets and the minor contaminating lymphocyte population were excluded by gating.

### 4.5. Cytokine Release

AML cells (1 × 10^6^ cells/mL in 1 mL/well) were seeded in 24-well plates with StemSpan SFEM^TM^ medium supplemented with the cytokines G-CSF, SCF, and FLT3-L and treated with one of the five SYK inhibitors. StemSpan SFEM^TM^ medium supplemented with cytokines G-CSF, SCF, and FLT3-L was treatment-free control. Supernatants were harvested after 48 h and stored at −80 °C until analyzed. Human Magnetic Luminex Assays (R&D Systems; Minneapolis MN, USA) were used to analyze all supernatant samples according to the manufacturer’s protocol. Cytokine release was estimated as a ratio to untreated control for each cytokine and SYK inhibitor.

### 4.6. Statistics and Bioinformatics

Graphs and figures were made using GraphPad Prism v. 5.02 (Graph Pad Software, San Diego, CA, USA). For statistical analysis, Fisher’s exact test was performed by using Stata 17 Software (StataCorp LLC, Lakeway, TX, USA) [63] and Wilcoxon signed-rank test or Mann–Whitney U test was performed using GraphPad Prism v. 5.02 (Graph Pad Software, San Diego, CA, USA). p-values < 0.05 were regarded as statistically significant. J-Express 2012 (Norwegian Bioinformatics Platform, Bergen, Norway) was used to perform cluster analysis. Flow cytometry data were analyzed using the FlowJo^TM^ v10.8.0 Software (BD Life Science, Franklin Lakes, NJ, USA) [62].

## 5. Conclusions

To conclude, our present results suggested that disrupting SYK signaling can alter the growth, viability, and mediator release of primary AML cells. Thus, SYK inhibitors can directly affect the leukemic cells and may also potentially, indirectly affect leukemogenesis. In the current study, we investigated five SYK inhibitors with different pharmacodynamic and pharmacokinetic properties. The specificity of the different inhibitors for SYK and other tyrosine kinases varies somewhat, and it is therefore not surprising that they had different effects in different AML patient samples. All the inhibitors, however, demonstrated significant concentration-dependent antiproliferative effects in our study. This study supports further investigation of the utilization of SYK inhibition in AML, and further preclinical, translational, and clinical studies should be performed to capture the features and benefits of SYK inhibition in AML.

## Figures and Tables

**Figure 1 ijms-23-14706-f001:**
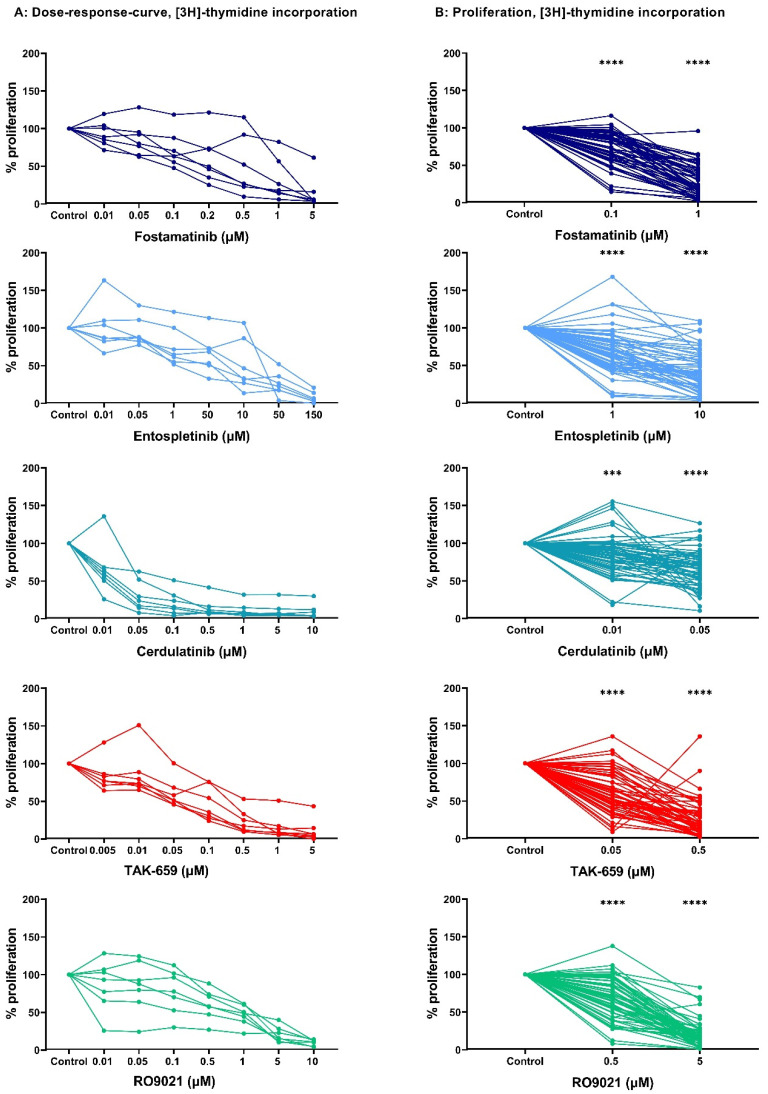
The antiproliferative effects of five different SYK inhibitors on primary human AML cells were examined using the [^3^H]-thymidine incorporation assay after seven days of treatment. The results are presented as the percent of untreated control (cells in medium alone) set to 100% on the y-axis. Each line represents one patient. (**A**) The proliferation of AML patient-derived cells from seven AML patients after in vitro treatment with seven different concentrations of each inhibitor. (**B**) AML cells derived from 59 patients that had detectable proliferation (>1000 cpm) were treated with two selected concentrations of each inhibitor, demonstrating the heterogeneity of the response to the SYK inhibitors. Overall, cell cultures treated with inhibitors showed a statistically significant reduction in AML cell proliferation compared to the untreated controls. The p-values of treated versus untreated controls are listed in the figure (*** *p* < 0.001, **** *p* < 0.0001, Wilcoxon signed-rank test).

**Figure 2 ijms-23-14706-f002:**
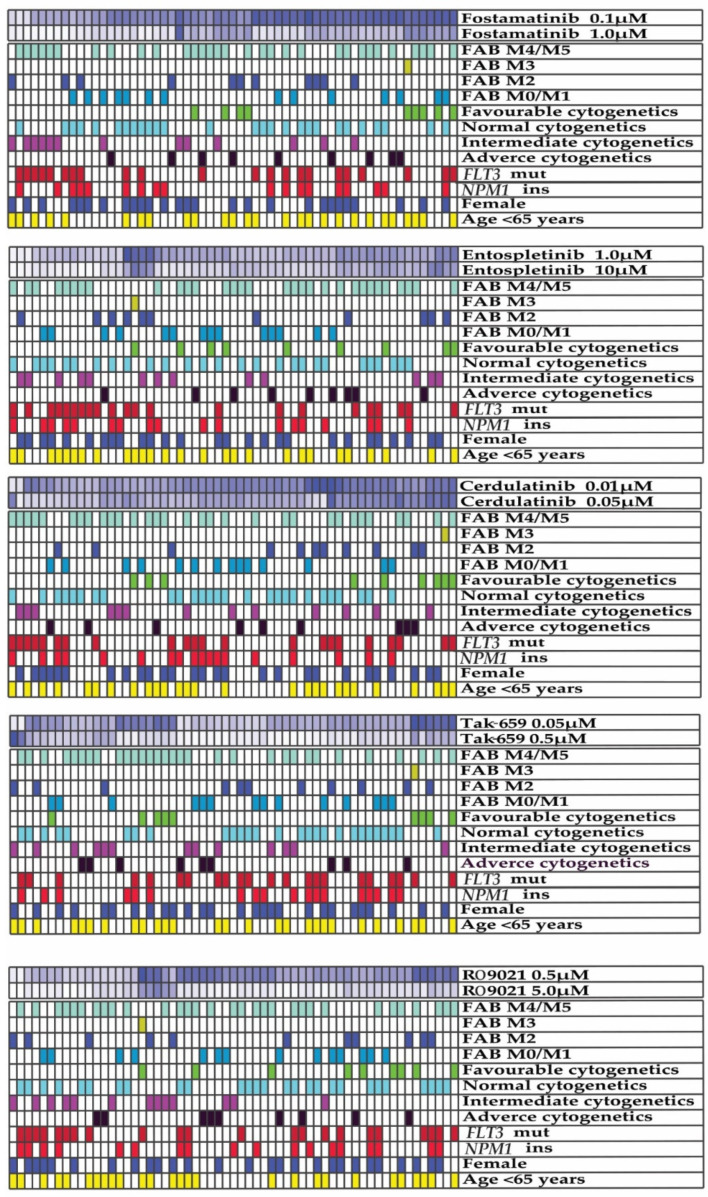
SYK inhibitors demonstrated antiproliferative effects across different AML entities. AML cells derived from 68 consecutive AML patients were cultured in medium and treated with SYK inhibitors or medium alone (untreated controls) for 7 days. AML cell proliferation was evaluated using the [^3^H]-thymidine incorporation assay, and 59 of the patient samples had detectable proliferation (defined as >1000 cpm). Patient samples were ranked based on the percentile of antiproliferative effects of the five different SYK inhibitors. Lower proliferative activity compared to controls, i.e., a stronger antiproliferative effect with pharmacological interventions, was marked by decreasing intensity of blue color. The darkest blue color demonstrated an antiproliferative effect corresponding to the 10th percentile, and the white color corresponded to the 90th percentile. Two patient cohorts were identified by classifying the overall antiproliferative effect as high or low based on the 50th percentile. Fisher’s exact test was used to evaluate the effect of etiology, FAB classification, cytogenetics, *FLT3* mutation, *NPM1* insertion, CD34 expression, gender, age, and SYK inhibition, and no significant associations were found. Although they demonstrated a heterogeneous picture, SYK inhibitors demonstrated antiproliferative effects across all these entities. Abbreviations: FAB represents French–American–British; *FLT3* represents FMS-like tyrosine kinase 3; and *NPM1* represents nucleophosmin 1.

**Figure 3 ijms-23-14706-f003:**
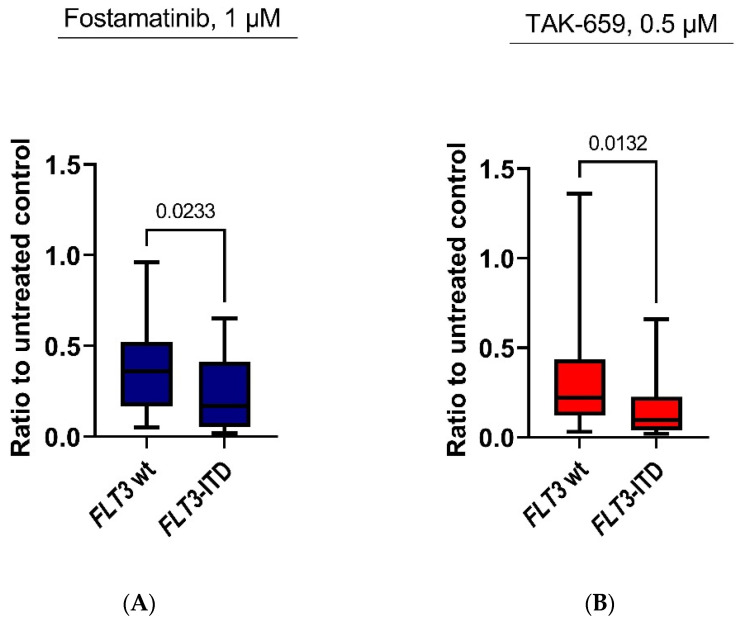
The antiproliferative response of AML primary cells treated with fostamatinib and TAK-659 based on *FLT3* mutation status assessed by the [^3^H]-thymidine proliferation assay. AML cells from 59 patients were cultured in medium and treated with SYK inhibitors or medium alone (untreated controls) for 7 days. For 3 of these 59 patients, the *FLT3*-mutation status was unknown. The ratio between the treated cultures versus untreated cultures (set to 1) was compared in patients with *FLT3*-wild type (wt) (34 patients) and *FLT3*-ITD (22 patients), and this ratio was significantly lower in *FLT3*-ITD patients after treatment with (**A**) 1 µM fostamatinib and (**B**) 0.5 µM TAK-659. Results are presented as the median levels with 25/75-percentiles (boxes) and with max to min values (whiskers). *p*-values are shown at the top of the figures (Mann–Whitney U test). Abbreviations: *FLT3* represents FMS-like tyrosine kinase 3.

**Figure 4 ijms-23-14706-f004:**
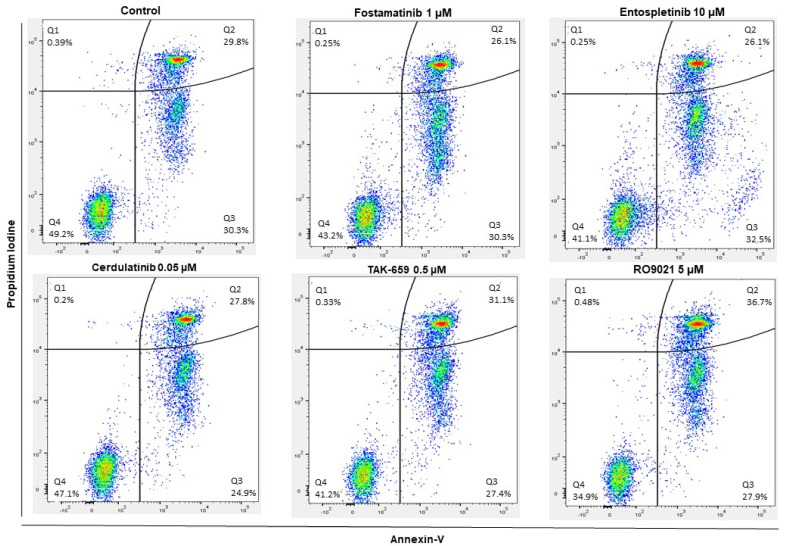
Example of gating strategy illustrated for one representative patient sample using the annexin-V/propidium iodine (PI) assay. The highest concentrations of each SYK inhibitor are demonstrated in their respective panels. The cells were cultivated with medium alone (upper left) or treated with one of the five SYK inhibitors for 48 h before samples were measured by flow cytometry using the Pacific Blue^TM^ Annexin V Apoptosis Detection Kit with PI. 10,000 events were collected and analyzed per sample. The plots show the percentage of viable, annexin V^−/^PI^−^ (square Q4); apoptotic, annexin V^+^/PI^−^ (square Q3); and necrotic cells, annexin V^+^/PI^+^ (square Q2).

**Figure 5 ijms-23-14706-f005:**
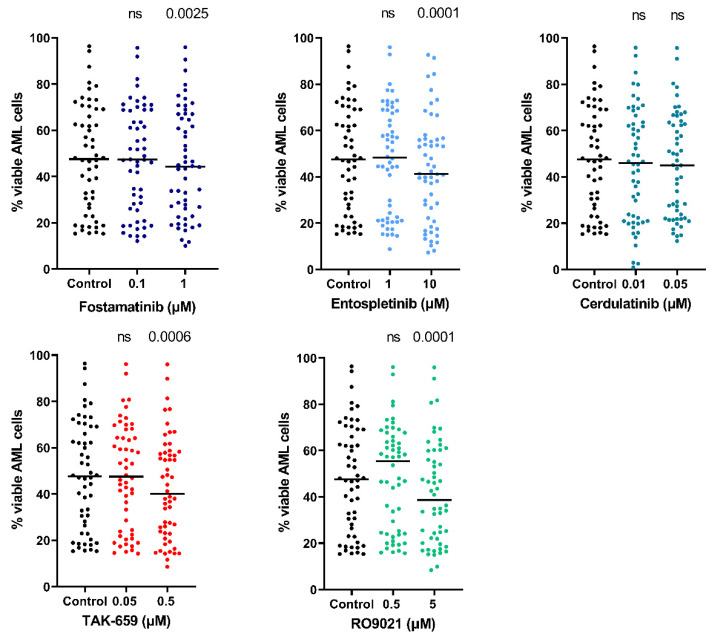
Reduced viability after SYK inhibition in AML primary cells. AML patient-derived cells were cultivated in medium alone or in the presence of SYK inhibitors. After 48 h of culture/treatment, viable cells and cell death were assessed by annexin-V/PI staining followed by flow cytometry analysis. Data were analyzed for the 54 patient samples with at least 15% viable cells in untreated cultures. Each dot represents results for one patient for viable cells in control cultures and intervention cultures. Solid lines indicate median percentage values. Wilcoxon signed-rank test was used for evaluation of treated samples against their untreated controls. Treatment with 1 µM fostamatinib, 10 µM entospletinib, 0.5 µM TAK-659, and 5 µM RO9021 significantly reduced AML cell viability. Abbreviations: AML represents acute myeloid leukemia; ns represents non-significant; and PI represents propidium iodine.

**Figure 6 ijms-23-14706-f006:**
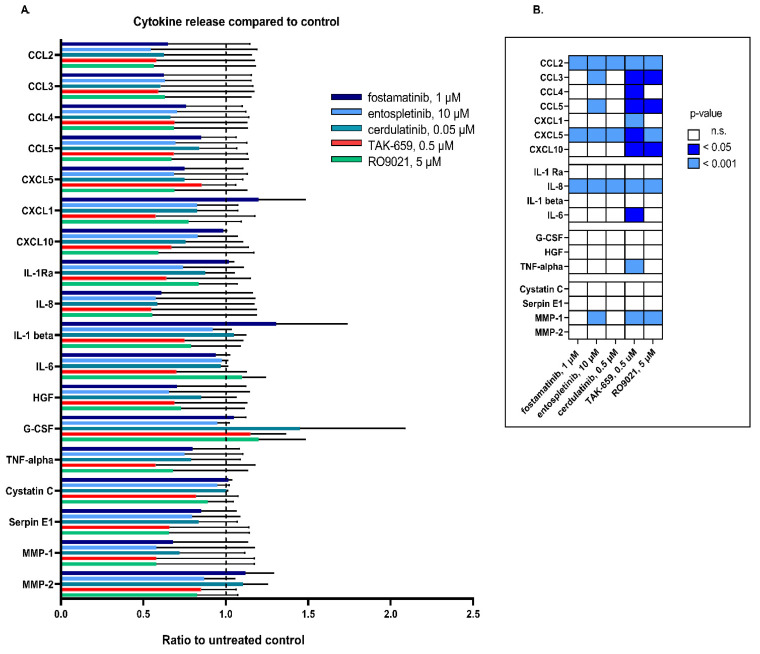
Effects of SYK inhibition on the cytokine release by primary AML cells. AML cells from 13 patients were cultured for 48 h in a medium with or without one of the five SYK inhibitors before the concentrations of cytokines were assessed in supernatants using multiplex analysis. (**A**) Shows the changes in the release of 18 cytokines after SYK inhibition. Results were presented as a ratio of median levels of cytokines measured in treated cultures compared to untreated control cultures, set to 1 (dotted line). Bars represent the median with standard deviation. The different SYK inhibitors are indicated by different colors. (**B**) Presents the resulting statistical analysis after SYK inhibition on the cytokine release. Wilcoxon signed-rank test was used to evaluate the treated patient samples against the untreated controls. Significant *p*-values are indicated by the color grading in the heatmap (*p* < 0.05–dark blue, *p* > 0.001–light blue, and not significant–white). Abbreviations: IL represents interleukin; Ra represents receptor antagonist; G-CSF represents granulocyte colony stimulating factor; HGF represents hepatocyte growth factor; TNF-alfa represents tumor necrosis factor alfa; and MMP represents matrix metalloproteinase.

**Figure 7 ijms-23-14706-f007:**
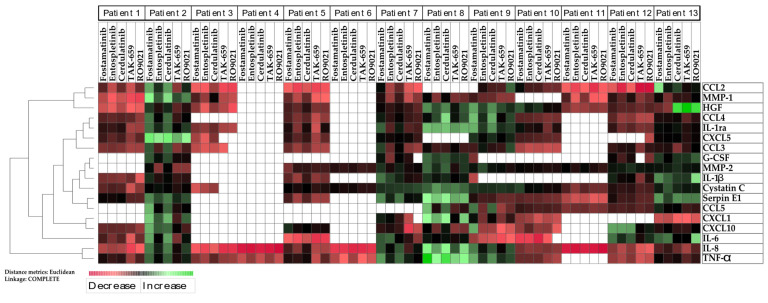
Heterogeneous effects of SYK inhibition on cytokine release by primary AML cells. Unsupervised hierarchical cluster analysis with heatmap shows the overall effect of the five SYK inhibitors on cytokine release in the 13 consecutive AML patients. AML cells were cultured for 48 h with or without SYK inhibitors before the concentrations of the cytokines were determined in supernatants using Luminex multiplex analysis. Results were presented as the ratio measured in treated cultures compared to untreated control cultures for each cytokine and inhibitor, and the heatmap demonstrated changes in the detectable release of cytokines using an increasing red color for decreasing levels of cytokine release and an increasing green color for increasing levels of cytokines. Black indicates no alteration of the release. White denotes no release of the cytokine for the given patient. Abbreviations: IL represents interleukin; Ra represents receptor antagonist; G-CSF represents granulocyte colony stimulating factor; HGF represents hepatocyte growth factor; TNF-α represents tumor necrosis factor alfa; and MMP represents matrix metalloproteinase.

**Table 1 ijms-23-14706-t001:** The overall median cell proliferative response of AML patient samples after treatment with two concentrations of each SYK inhibitor. Results of 59 AML patient samples with proliferation > 1000 cpm, assessed by the [^3^H]-thymidine incorporation assay, are presented as percent proliferation compared to untreated controls.

Inhibitor, Concentration	Median Percent Proliferation Compared to Control (Range)
Fostamatinib, 1 µM Fostamatinib, 0.1 µM	27% (2–96) 74% (14–116)
Entospletinib, 10 µM Entospletinib, 1 µM	36% (4–109) 65% (9–101)
Cerdulatinib, 0.05 µM Cerdulatinib, 0.01 µM	66% (10–126) 86% (18–105)
TAK-659, 0.5 µM TAK-659, 0.05 µM	19% (2–112) 56% (9–103)
RO9021, 5 µM RO9021, 0.5 µM	20% (3–88) 66% (8–103)

**Table 2 ijms-23-14706-t002:** Cytokine release from primary AML cells. AML cells from 13 patients were cultured for 48 h before supernatants were analyzed using human magnetic Luminex multiplex assays. The number of patients with detectable release in untreated controls, the median level, and the variation range for each of the eighteen cytokines are shown. IL represents interleukin; Ra represents receptor antagonist; G-CSF represents granulocyte colony stimulating factor; HGF represents hepatocyte growth factor; TNF-α represents tumor necrosis factor alfa; and MMP represents matrix metalloproteinase.

Cytokine	Number of Patients with Detectable Release	Median Levels (pg/mL)	Variation Range (pg/mL)
Chemokines			
CCL 2	8	1478	35.1–10,521
CCL3	11	1745	75–17,933
CCL4	11	980	91–12,622
CCL5	10	106	6.0–675
CXCL1	7	1007	59–6965
CXCL5	10	1064	178–16,771
CXCL10	11	8.9	0.2–290
Interleukins			
IL-1 Ra	11	2453	134–29,599
IL-8	12	11,266	164–111,476
IL-1β	9	24.6	6.4–458
IL-6	9	55	8.9–956
Growth factors			
G-CSF	8	28.1	5.6–141
HGF	11	65	14–806
TNF-α	13	11.6	1–495
Protease system			
Cystatin C	12	5302	1379–13,223
Serpin-E1	10	370	37.2–12,243
MMP-1	11	139	19.5–519
MMP-2	10	2808	1079–8714

Abbreviations: IL represents interleukin; Ra, receptor antagonist; G-CSF, granulocyte colony stimulating factor; HGF, hepatocyte growth factor; TNF-α, tumor necrosis factor alfa; MMP, matrix metalloproteinase.

**Table 3 ijms-23-14706-t003:** Patient characteristics. Demographical and clinical characteristics of the 68 acute myeloid leukemia (AML) 30% patients included in the study.

Patient Characteristics		Observations
Demographic Data and Disease History		
Gender (number, percent)	Female/male	31/37 (46/54)
Age, years (median, range)		62 (17–87)
Disease History (number, percent)	De novo	51 (75)
	Secondary	15 (22)
	Relapse	2 (3)
Hematology (mean, range)		
Hemoglobin, g/dL		9.9 (6.3–15.4)
Platelets, ×10^9^/L		77 (5–258)
AML cell differentiation (number, percent)		
FAB	M0-1	13 (19)
	M2	13 (19)
	M3	1 (2)
	M4-5	38 (56)
CD34	Unclassified Positive (≥30%) Negative (≤30%) Unknown	3 (4) 37 (54) 27 (40) 4 (6)
Cytogenetic and genetic abnormalities (number, percent)		
Cytogenetics *	Favorable	9 (13)
	Intermediate	15 (22)
	Normal	31 (46)
	Adverse	11 (16)
	Unknown	2 (3)
*FLT3*	Wild type	39 (57)
	ITD TKD	24 (37) 1 (2)
	Unknown	4 (6)
*NPM1*	Wild type	44 (65)
	Insertion	22 (32)
	Unknown	2 (3)

Abbreviations: FAB represents French–American–British; *FLT3* represents FMS-like tyrosine kinase 3; ITD represents internal tandem duplication; *NPM1* represents nucleophosmin 1; and TKD represents tyrosine kinase domain. * Cytogenetic risk classification is based on the European Leukemia Net Classification of 2022 [1].

**Table 4 ijms-23-14706-t004:** Short description of the five SYK inhibitors used in our study.

Inhibitor	Molecular Weight	Pharmacological Properties	Company	Chemical Structure	Cas Number
Fostamatinib Disodium (R788)	624.42	Oral prodrug of the active compound R406. Competitive SYK/FLT3 inhibitor. Demonstrated in vitro antiproliferative effect in concentrations of 0.8–8.1 µM to DLBCL [23] and 41 nM to CLL [24].	MedChemExpress (Monmouth Junction, NJ, USA)	C_23_H_24_FN_6_Na_2_O_9_P	1025687-58-4
Entospletinib (GS-9973)	411.46	Orally bioavailable, selective SYK inhibitor. Induced antiproliferative and proapoptotic effects in vitro in pre-B-ALL and pro-B-ALL cell lines with concentrations of 0.001–20 µM [22].	MedChemExpress (Monmouth Junction, NJ, USA)	C_23_H_21_N_7_O	1229208-44-9
Cerdulatinib (PRT062070; PRT2070)	482	Orally active, multi-targeted tyrosine kinase inhibitor to JAK1/JAK2/JAK3/TYK2 and SYK. Induced apoptosis of CLL cells in vitro with concentrations of 0.3–1 µM [25], suppressed cell proliferation, and reduced cell viability in T-cell lines with a concentration of 10 µM [26].	Selleckchem (Houston, TX, USA)	C_20_H_27_N_7_O_3_S	1198300-79-6
TAK-659	380.85	Orally bioavailable, selective SYK inhibitor. Selective against most other kinases, but potent toward SYK and FLT3. Induced in vitro apoptosis of CLL cells in concentrations of 0.1–10 µM [27].	Selleckchem (Houston, TX, USA)	C_17_H_21_FN_6_HCl	1952251-28-3
RO9021	355.44	Potent SYK inhibitor. Suppresses BCR signaling and B-cell proliferation at 1 µM [28].	Selleckchem (Houston, TX, USA)	C_18_H_25_N_7_O	1446790-62-0

## Data Availability

Not applicable.

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
