# Peer review of "Heterogeneity of Patient-Derived Acute Myeloid Leukemia Cells Subjected to SYK In Vitro Inhibition"

_ijms, 2022, doi:10.3390/ijms232314706_

Round 1

Reviewer 1 Report

MK Brattås et al. present heterogeneity of patient-derived acute myeloid leukemia cells through syk cascade inhibition in vitro.

This article is well organized, but the reviewer has some concerns.

Recently, therapies such as venetclax, a BCL inhibitor, and gemutuzumab ozogamicin, a CD33 therapeutic agent, have greatly changed the therapeutic system of leukemias. Please comment this clinical progress in Discussion.

Although FLT3 mutations have been repeatedly discussed, the fact that there is only one patient with TKD mutations (2%) is a very small impression compared to previous reports. It is necessary to examine whether these results indicate that mutation analysis has not been performed on TKD, or study limitations with small group sample collections.

The FLT3 inhibitors midostaurin, quizartinib, and gilitertinib have clearly improved the prognosis of FLT3 mutation-positive acute leukemias.

There may be discussed as to whether Syk inhibitors may be useful when resistance to FLT3 inhibitors develops.

Minor

The expressions for NPM1 insertion and mutation should be united.

Author Response

MK Brattås et al. present heterogeneity of patient-derived acute myeloid leukemia cells through syk cascade inhibition in vitro.

This article is well organized, but the reviewer has some concerns.

We are grateful for the mainly positive comments from the reviver, and the constructive criticism, which indeed has helped us improved the manuscript.

Recently, therapies such as venetoclax, a BCL inhibitor, and gemutuzumab ozogamicin, a CD33 therapeutic agent, have greatly changed the therapeutic system of leukemias. Please comment this clinical progress in Discussion.

We recognize the evolving specters of pharmacological agents currently entering the armamentarium for AML therapies. We have improved the discussion of this new approaches in our article, and also adding in the discussion potential synergistic effect these inhibitors and SYK inhibitors can have in AML

Although FLT3 mutations have been repeatedly discussed, the fact that there is only one patient with TKD mutations (2%) is a very small impression compared to previous reports. It is necessary to examine whether these results indicate that mutation analysis has not been performed on TKD, or study limitations with small group sample collections.

The patients in our study have been evaluated for both the FLT3-ITD and FLT3-TKD mutation. We agree that the prevalence of FLT3-TKD mutation is a little lower than expected. However, the prevalence of FLT3-TKD mutation in AML is predicted to be not more than approximately 5%. Hence in our limited cohort it is not surprisingly that only one patient had this mutation. We have shortly discussed the difference in FLT3.

The FLT3 inhibitors midostaurin, quizartinib, and gilitertinib have clearly improved the prognosis of FLT3 mutation-positive acute leukemias.

There may be discussed as to whether Syk inhibitors may be useful when resistance to FLT3 inhibitors develops.

We also recognize the introduction of new and more selective FLT3 inhibitors, and their impact of further AML therapy. Accordingly, we have discussed these features also in our revised version of the manuscript.

Reviewer 2 Report

How many patient samples were CD34+?

The authors should make use of the recent publication from John Dick's group on an AML cell hierarchy where they classify leukemia cell samples as Primitive, GMP-like, or mature, to classify their patient samples. Can the authors comment on the cellular composition of the patient samples to their response to SYK inhibitors?

Can the authors comments on the specificity of SYK inhibitors in leukemia cells vs normal hematopoietic cells?

Author Response

How many patient samples were CD34+?

We are grateful for this comment regarding CD34+ cells, as this is an important marker in AML cell biology. This data was obtained by multiparametric flow cytometry analysis at the time of diagnosis, and registered. Accordingly, we have included CD34 positive features for our AML patient cohort in the revised version of the manuscript. Furthermore, we also performed an analysis to see if CD34 positive cells were more vulnerable for SYK inhibition.

The authors should make use of the recent publication from John Dick's group on an AML cell hierarchy where they classify leukemia cell samples as Primitive, GMP-like, or mature, to classify their patient samples. Can the authors comment on the cellular composition of the patient samples to their response to SYK inhibitors?

We are grateful for the comment regarding this paper, and the proposed classification system as primitive, mature, GMP-like and intermediate. Unfortunately, we are lacking the baseline data needed to perform this classification, and this is a new feature, which must be further validated in AML. However, we have discussed these features of AML classification in light of this new paper and have discussed the future horizon for including such data, in the setting of SYK inhibition in AML.

Can the authors comments on the specificity of SYK inhibitors in leukemia cells vs normal hematopoietic cells?

This is an important question, and our studies has not particularly focused on potential differences between SYK inhibitors in leukemic cells and in normal hematopoietic cells. However, the have discussed these features in the revised version of our manuscript, and how future studies also should evaluate this aspect of SYK inhibition. 

Reviewer 3 Report

The authors have tested 5 SYK inhibitors in an in vitro primary AML cell culture system in an attempt to determine effects on proliferation and apoptosis. An impressive number of samples were used but given the heterogenous response among the samples it is hard to be convinced of any concrete findings. There is a lack of non-malignant control cells for the experiments. Not surprisingly, many of the primary AML samples do not survive or proliferate well in vitro. A better approach would be to focus on fewer samples that do adapt well to culture and perform more detailed analysis on those samples. The data shown in aggregate is not convincing and seems to mask the inadequacy of the experimental design. There is a focus on inhibition of cytokine release from the cells, but it is not clear to me that this is consequential.

It is surprising that the data in Fig 1A from 7 separate samples creates lines with fairly small error bars. This seems to not reflect the heterogeneity of the larger 58 sample panel in Fig1B. It would be better to show all 7 samples as individual lines on each chart in Fig1A.

Table 1. Shouldn’t “Median percent inhibition” be “Median % Proliferation”?

A normal BM or other suitable non-malignant control should be used to compare to the AML proliferation response in the culture conditions used. Are the compounds specific for AML in the culture conditions used here?

It is possible that the addition of high level of Flt3-L in the culture media is minimizing or masking an increased effect of SYK inhibitors on FLT-3 mutant AMLs. The authors should consider alternative cytokine conditions (much lower FL or no FL).

Figure 4 should show representative control and treated plots from the same sample. The authors use 5% viability in the controls non-treated sample as a cut off for acceptability. This seems far too low. It is hard to interpret the effects of the compounds on cell cultures that are of such low viability. Have the authors verified that the few surviving and proliferating cells in these low viability samples are actually blasts and not residual normal cells?

Figure 5 is not convincing. Any effects are very subtle for high doses. A and B are essentially the same data inverted (mirror image).

It seems that there should be error bars for Figure 6A if the data is from 13 patient samples. Control cells should be included in this experiment.

I’m not sure what Figure 7 adds over Figure 6.

Author Response

The authors have tested 5 SYK inhibitors in an in vitro primary AML cell culture system in an attempt to determine effects on proliferation and apoptosis. An impressive number of samples were used but given the heterogenous response among the samples it is hard to be convinced of any concrete findings. There is a lack of non-malignant control cells for the experiments. Not surprisingly, many of the primary AML samples do not survive or proliferate well in vitro. A better approach would be to focus on fewer samples that do adapt well to culture and perform more detailed analysis on those samples. The data shown in aggregate is not convincing and seems to mask the inadequacy of the experimental design. There is a focus on inhibition of cytokine release from the cells, but it is not clear to me that this is consequential.

 It is surprising that the data in Fig 1A from 7 separate samples creates lines with fairly small error bars. This seems to not reflect the heterogeneity of the larger 58 sample panel in Fig1B. It would be better to show all 7 samples as individual lines on each chart in Fig1A.

As the initial experiment was performed on a broader range of concentration, we believe that it is not very surprising that the highest concentrations show stronger antiproliferative effects, and the lower concentrations demonstrate no or little antiproliferative effects in all patients. To better demonstrate this, we have as suggested from the reviewer demonstrated lines for each individual patients in the revised version of figure 1.

Table 1. Shouldn’t “Median percent inhibition” be “Median % Proliferation”?

This is correct, and accordingly we have altered this in our revised version

 A normal BM or other suitable non-malignant control should be used to compare to the AML proliferation response in the culture conditions used. Are the compounds specific for AML in the culture conditions used here?

We agree that normal bone marrow cells also could be used as controls. However, as leukemic blasts are clearly distinguished from for example peripheral mononuclear cells ((PMCS), we believe this has limited important for our study. The main aims of our present study were to investigate differences within the AML patient cohort, and hence not primarily compared to heathy bone marrow. Furthermore, SYK seem to have little myelosuppressive effect in vivo study, and hence should be tolerated. We have discussed these important features in the revised version of our manuscript.

It is possible that the addition of high level of Flt3-L in the culture media is minimizing or masking an increased effect of SYK inhibitors on FLT-3 mutant AMLs. The authors should consider alternative cytokine conditions (much lower FL or no FL).

We agree in the principle that FLT3-L could affected the culturing conditions and the media performed. However, FLT3-L is often used in standardized assay for investigation of leukemic cells in vitro and is regarded as important/essential for stimulating proliferation of leukemic blasts. We have however discussed these features in our revised version of the manuscript.

Figure 4 should show representative control and treated plots from the same sample. The authors use 5% viability in the controls non-treated sample as a cut off for acceptability. This seems far too low. It is hard to interpret the effects of the compounds on cell cultures that are of such low viability. Have the authors verified that the few surviving and proliferating cells in these low viability samples are actually blasts and not residual normal cells?

Figure 5 is not convincing. Any effects are very subtle for high doses. A and B are essentially the same data inverted (mirror image).

By density gradient separation we have demonstrated that the leukemic cells consist of >95 % of cell population, and furthermore by forward-side scatter we have eliminated debris and other cell population not consisting of blasts. We agree that is also suitable to present a figure of intervention cultures, and accordingly we have performed this in our revised version. Spontaneous apoptosis will occur within the leukemic blasts by in vitro conditioning; however, we agree that only 5% is, and hence we have increase this to 15% in our revised version of the manuscript. Figure 5 is hence revised and only panel A is kept.

It seems that there should be error bars for Figure 6A if the data is from 13 patient samples. Control cells should be included in this experiment.

We agree that error bars should be included, and accordingly this is added in the revised version of the figure. The data is presented as ratio of control, and hence control cultures in itself is not presented, although we data are described in Figure 2.

 I’m not sure what Figure 7 adds over Figure 6.

Although we agree that some of the same data is present in Figure 6 and Figure 7, we will argue that Figure 7 demonstrated the heterogeneity in the patient material, and the general class effects of SYK inhibitors in a more proper way than figure 1, and hence we will argue for keeping Figure 7 in the manus.

Round 2

Reviewer 2 Report

CD34 has been written as CD43. Please correct.

Other than that, the authors have responded to the comments to satisfaction. No further comments.

Author Response

We are very grateful for these comments. Minor spell check is performed, and typos corrected. 

Reviewer 3 Report

The manuscript has slightly improved, however the previous concerns remain and should be addressed more thoroughly.

Author Response

Regarding the previous review report, we tried in review round 1 to make a thorough assessment, and performed significant alterations in our manuscript as suggested. It is a bit unclear what changes are desired now. However, we have again tried to make a change related to the discussion about the microenvironment and FLT3-L in our in vitro studies. We think this further has improved our manuscript.